# NAD^+^ Anabolism Disturbance Causes Glomerular Mesangial Cell Injury in Diabetic Nephropathy

**DOI:** 10.3390/ijms23073458

**Published:** 2022-03-22

**Authors:** Xue Li, Yankun Li, Fengxia Li, Qi Chen, Zhonghua Zhao, Xueguang Liu, Nong Zhang, Hui Li

**Affiliations:** Department of Pathology, School of Basic Medical Sciences, Fudan University, Shanghai 200032, China; 18111010074@fudan.edu.cn (X.L.); 19111010070@fudan.edu.cn (Y.L.); 20111010074@fudan.edu.cn (F.L.); 13381620680@163.com (Q.C.); 199752@sina.com (Z.Z.); glxg69@shmu.edu.cn (X.L.); nzhang@fudan.edu.cn (N.Z.)

**Keywords:** Sirt1, NAD^+^, NMN, diabetic nephropathy, NMNAT1

## Abstract

The homeostasis of NAD^+^ anabolism is indispensable for maintaining the NAD^+^ pool. In mammals, the mainly synthetic pathway of NAD^+^ is the salvage synthesis, a reaction catalyzed by nicotinamide mononucleotide adenylyltransferase (NAMPT) and nicotinamide mononucleotide adenylyltransferase (NMNATs) successively, converting nicotinamide (NAM) to nicotinamide mononucleotide (NMN) and NMN to NAD^+^, respectively. However, the relationship between NAD^+^ anabolism disturbance and diabetic nephropathy (DN) remains elusive. Here our study found that the disruption of NAD^+^ anabolism homeostasis caused an elevation in both oxidative stress and fibronectin expression, along with a decrease in Sirt1 and an increase in both NF-κB P65 expression and acetylation, culminating in extracellular matrix deposition and globular fibrosis in DN. More importantly, through constitutively overexpressing NMNAT1 or NAMPT in human mesangial cells, we revealed NAD^+^ levels altered inversely with NMN levels in the context of DN and, further, their changes affect Sirt1/NF-κB P65, thus playing a crucial role in the pathogenesis of DN. Accordingly, FK866, a NAMPT inhibitor, and quercetin, a Sirt1 agonist, have favorable effects on the maintenance of NAD^+^ homeostasis and renal function in db/db mice. Collectively, our findings suggest that NMN accumulation may provide a causal link between NAD^+^ anabolism disturbance and diabetic nephropathy (DN) as well as a promising therapeutic target for DN treatment.

## 1. Introduction

Diabetes mellitus (DM) is one of the fastest-growing diseases in the world, and is expected to affect nearly 700 million adults by 2045 [1]. Diabetic nephropathy (DN) occurs in patients with DM and is the foremost cause of end-stage renal disease globally, which accounts for a significant increase in mortality in these patients and poses a grave threat to the clinical outcome of diabetic patients [2]. It is necessary to properly assess the patient’s diabetic nephropathy and to formulate appropriate treatment guidelines and supervision [3]. Changes in intracellular metabolism, such as reactive oxygen species (ROS), hypoxia, and accumulation of advanced glycation end products (AGEs) plays a crucial role in pathological development of DN [4].

Nicotinamide adenine dinucleotide (NAD^+^) is a coenzyme for redox reactions, making it central to energy metabolism and an essential cofactor for non-redox NAD^+^-dependent enzymes, including sirtuin, CD38, and poly (ADP-ribose) polymerases [5,6]. NAD^+^ can be synthesized from three pathways, including the Preiss–Handler, salvage pathway, and de novo pathway. In mammals, the main synthetic pathway for NAD^+^ is the salvage pathway, whereby the precursors including nicotinamide (NAM), nicotinic acid (NA), and nicotinamide riboside (NR) are converted into an intermediate called nicotinamide mononucleotide (NMN), which is dependent on a rate-limiting enzyme, nicotinamide phosphoribosyl-transferase (NAMPT) [7]. The intermediate NMN can be converted into NAD^+^ via nicotinamide mononucleotide adenylyltransferase (NMNATs). NAD^+^ generated by this pathway is consumed by multiple enzymes including sirtuin, PARPs, and cADPR synthases, to generate NAM, which can be recycled in the salvage pathway. Despite numerous mechanisms underlying the pathogenesis of DN, glomerular injury driven by NAD^+^ homeostasis imbalance has been thought to be a pivotal pathophysiological mechanism of DN [8,9] with decreased cellular NAD^+^ concentrations and NAD^+^/NADH ratio being closely related to the pathogenesis of DN [10,11]. Mesangial cells (MSC) are critical for maintaining and regulating glomerular filtration, and the dysregulation of MSC secretion causes the accumulation of the mesangial extracellular matrix (ECM), among which fibronectin (FN) is putative as the most obviously impacted protein [12]. Therefore, in the environment of DN, excessive FN secretion by mesangial cells needs to be further studied. However, the effects that NAD^+^-related enzymes have on mesangial cells (MSC) under DN involving abnormally secreted FN, remain largely unknown.

Sirt1, a nicotinamide adenine dinucleotide (NAD^+^)-dependent protein deacetylase, a member of the sirtuin family, has been extensively studied for its role in diabetic complications. A number of models of diabetes have been used to evaluate the renoprotective effects of Sirt1, including both T1DM and T2DM models [13,14,15,16]. NF-κB is a transcription factor that governs the expression of genes involved in inflammation [17]. It is now known that Sirt1 can deacetylate the P65 subunit of NF-κB and inhibit NF-κB pro-inflammatory signaling and the production of FN in DN [18]. However, whether and to what extent NAD^+^ metabolic disorder can affect Sirt1/ NF-κB P65/FN remains to be further studied for better understanding of the determinants of Sirt1 beneficial effects in DN.

In this study, with in vitro and in vivo models of DN, we determined whether NAD anabolism disturbance exists, whether normalizing NAD anabolism or replenishing NAD pool can alleviate MSC injury, and how NMN, as an intermediate of NAD salvage synthesis, affects Sirt1/NF-κB P65/FN in the MSC. Addressing these questions contributes to shedding light on the importance of NAD^+^ anabolism disturbance involved in DN and on the exploration of new treatment target.

## 2. Results

### 2.1. NAD^+^-Related Metabolic Enzymes Alters in High Glucose Conditionally Injuried Mesangial Cells

To detect the changes in NAD^+^-related metabolic enzymes of MSC in high glucose environment, the expression of NAD^+^-related metabolic enzymes was determined by Western Blot after 0 to 6 days incubation with high glucose (HG) medium (Figure 1A). As shown in Supplementary Figure A, NMNAT1, a subtype of NMNATs for the final step of NAD^+^ synthesis, was the fundamental expression subtype in the glomerulus rather than NMNAT2 or NMNAT3. Therefore, NMNAT1 was used as the readout for NMNAT expression in MSC thereafter. Immunoblotting detection of expression of NAPRT (nicotinate phosphoribosyltransferase), an intracellular enzyme to catalyze the first step of the Preiss–Handler pathway, and NMNAT1 both showed time-dependent decreasing, whereas NAMPT increased over time under high glucose (Figure 1A). Sirt1, an NAD+-dependent deacetylase, reduced over time in response to high glucose (Figure 1A). Expectedly FN, a fibrosis maker of glomerulus, increased with time of high glucose culturing (Figure 1A). Furthermore, the HG group presented higher FN adhesion than the normal glucose (NG) group in the cell attachment assay (Figure 1B). Likewise, the DCFH-DA method was used to investigate the ROS levels, which showed a remarkable increase after 24 h HG incubation, suggesting the occurrence of oxidative stress in MSC (Figure 1C). Given the results of these experiments above, we confirmed that HG causes a disorder in NAD^+^ anabolism, namely a decrease in NMNAT1 expression and an increase in NAMPT expression in salvage pathway, and these alterations were accompanied with cellular damage, including oxidative stress, highly expressing of FN and cell adhesion.

### 2.2. NMN Level Parallels Mesangial Cell Injury When NMNAT1 or NAMPT Overexpressed

As we observed a decrease in the expression of NMNAT1 and an increase in the expression of NAMPT in a high glucose environment, we hypothesized that changes in the expression of NMNAT1 or NAMPT might have an effect on mesangial cell injury in an HG environment, and also suspected the changes in NMN, an intermediate metabolite or a synthetic precursor of NAD^+^, and its possible detrimental effects on cells. In order to prove the hypothesis, firstly, we successfully transfected MSC with a lentivirus vector encoding the constitutively NMNAT1 or NAMPT, which were corroborated by the immunoblotting of a markedly increased protein expression compared with the no-load vector transfected control (Figure 2A,B). Secondly, immunoblotting revealed a decrease in FN and NAMPT as well as an increase in Sirt1 in NMNAT1-overexpressed MSC under HG (Figure 2A). In contrast, NAMPT-overexpressed MSC showed the opposite results, that is decreased NMNAT1 and Sirt1 levels as well as increased FN expression (Figure 2B).

The DCFH-DA fluorescence intensity greatly reduced (Figure 2C) and the cell adhesion receded in NMNAT1-overexpressed group regardless of NG or HG condition (Figure 2E). However, NAMPT-overexpressed group exhibited a converse result. ROS and FN adhesiveness enhanced after overexpression of NAMPT (Figure 2D,F). LCMS assay measured a decrease in NAD^+^ and NAD^+^/NADH under HG but an increase in NMN, the product of NAMPT and substrate of NMNAT once NMNAT1 degraded (Figure 2G–J). These high glucose-induced changes were reversed by overexpression of NMNAT1 in MSC (Figure 2G–J). Similar to transfection of NMNAT1, overexpressed NAMPT group showed increasing level of NAD^+^ and NAD^+^/NADH in (Figure 2K–M) in LCMS assay. However, the NMN altered contrarily in NMNAT1-overexpressed and NAMPT-overexpressed groups compared to its accumulation in HG (Figure 2N). Together, all these data illustrated that the NMN level is a crucial determinant of high glucose-mediated injury in MSC since suppression of high glucose-induced NMN accumulation by NMNAT1 overexpression and promotion of high glucose-induced NMN accumulation by NAMPT overexpression brings MSC benefits and harms, respectively.

### 2.3. FK866 and Quercetin Alleviates NAD^+^ Anabolism Disturbance in HG-Injured MSC

Based on the salutary effects of NMNAT1 and the deleterious effects of NAMPT or NMN described above, we treated MSC with FK866 (NAMPT inhibitor) and quercetin (Sirt1 activator) under HG, respectively, to further test whether prevention of NMN accumulation or activation of Sirt1 can improve NAD^+^ anabolism disturbance and injury simultaneously in HG-injured MSC. Immunoblotting showed that the expression levels of Sirt1, NMNAT1, and NAPRT were markedly increased while FN and NAMPT levels decreased in HG by FK866 or quercetin treatment for 24 h (Figure 3A). The levels of NAD^+^ and NADH also were consistently increased by FK866 or quercetin treatment in HG (Figure 3B–D). In addition, both treatments reduced accumulation of NMN and NADH in HG (Figure 3E). Furthermore, we examined the effects of FK866 and quercetin on glycolysis and redox homeostasis under HG stress with LCMS (Figure 3F–K). The results showed that GSH (Figure 3J) and NADPH (Figure 3F) decreased, accompanied by an increase in GSSG (Figure 3I) and NADP^+^ (Figure 3G) under HG. FK866 or quercetin treatment increased GSH/GSSG (Figure 3K) and GSH (Figure 3J) but decreased GSSG (Figure 3I) and NADP^+^ (Figure 3G) compared with the HG group. These results indicated that restoring NMN or activating Sirt1 mitigates glycolysis and FN secretion of MSC under HG.

### 2.4. FK866 and Quercetin Improves Insulin Resistance and Kidney Function in db/db DN Mice

To validate the effect of restoring NMN or activating Sirt1 in vivo in DN, male C57BL/6J and db/db mice (7 to 9 weeks old) were used and divided into four groups: wild type (C57BL/6J), db/db, db/db with FK866 (20 mg/kg, biw, i.p.), or db/db with quercetin (100 mg/kg, biw, i.p.). After four weeks of medication during 16 to 20 weeks old, all the mice were sacrificed at 20 weeks to obtain kidney tissue, plasma, and urine, then metabolic indices associated with diabetes were measured. The weight (Figure 4B), blood glucose (Figure 4D), and serum insulin (Figure 4E) of those mice were significantly increased in db/db, but kidney weight to body weight ratio (Figure 4C) was lower than wild type mice. Renal function parameters measured in the study, including blood urea nitrogen (Figure 4F), serum creatinine (Figure 4G), and urine albumin to urine creatinine ratio (UACR) (Figure 4H) were significantly increased in diabetic db/db mice compared with WT. Treatment with FK866 or quercetin in diabetic mice did not change their weights (Figure 4B), kidney weight to body weight ratio (Figure 4C) or blood glucose (Figure 4D). However, treated with FK866 or quercetin markedly blunted the increase in serum insulin (Figure 4E), blood urea nitrogen (Figure 4F), serum creatinine (Figure 4G), and UACR (Figure 4H) in diabetic db/db mice. Consistently, matrix expansion and glycogen accumulation assessed by Periodic Acid–Schiff (PAS), Masson, and HE staining presented in the mesangial region of non-treated db/db mice, but not in FK866-treated or quercetin-treated db/db mice or wild type mice (Figure 4I). In addition, the elevated glomerular sclerosis index (GSI) shown in DN group decreased in both treatment groups (Figure 4J). To summarize, treatment of inhibiting NAMPT or activating Sirt1 was able to suppress insulin resistance and renal injury as evidence by improved physiological and histopathological parameters in db/db mice.

### 2.5. FK866 and Quercetin Lessens NAD^+^ Anabolism Disturbance and FN Overproduction in db/db DN Mice

We logically hypothesized and examined whether FK866 and quercetin played a role in lessening NAD^+^ anabolism disorder in vivo. NAD^+^-related metabolic enzymes were determined by immunohistochemical staining of renal biopsies (Figure 5A) and immunoblot analysis of renal tissue protein (Figure 5B) from each experimental group mice. db/db mice showed decreases in NMNAT1, Sirt1, and NAPRT levels but increases in NAMPT and FN expression, suggesting the existence of NAD^+^ anabolism disturbance and glomerular matrix overproduction in DN. Levels of NAD^+^ (Figure 5C), NADH (Figure 5D), and the NAD^+^/NADH (Figure 5E) ratio in mice renal tissues were evaluated using commercial kit. Although there was no significant change in NADH (Figure 5D), NAD^+^ levels (Figure 5C) and the values of NAD^+^/NADH (Figure 5E) ratio notably reduced in the renal tissue from the DN group compared with the WT group. Consistent with results in vitro above, the FK866 or quercetin treatment restored expression of NAD^+^ and NAD^+^/NADH compared with the non-treated counterparts. These findings confirmed that neither NAMPT suppression nor Sirt1 activation could efficaciously mitigate the NAD^+^ anabolism disturbance and the fibrosis maker in DN.

### 2.6. NMN Accumulation Causes Changes in Sirt1/NF-κB P65/FN Associated with DN

Sitr1 deficiency is a well-recognized contributor to extracellular matrix deposition of fibrosis and oxidative stress in DN. NF-κB P65 has been confirmed as a downstream target of Sirt1 deacetylation and some studies have proposed that Sirt1/ NF-κB P65 is correlated with diabetic renal fibrosis. We examined if NMN accumulation-induced renal cells damage would link to Sirt1/NF-κB P65. As shown in Figure 6A, immunohistochemical staining presented an increase in P65 expression and acetylated P65/P65 expression ratio in the db/db DN mice group compared with the wild type group. Incubated with 0 to 0.5 mM NMN for 24 h, MSC decreased Sirt1 expression, whereas increased FN expression in a dose dependent manner (Figure 6B). To further explore the impact of NMN on Sirt1 expression in HG-induced MSC injury, MSC were exposed to a 24 h incubation of NMN (0.5 mM) or not. The exposure resulted in a significantly low expression of Sirt1 but an evidently high expression of FN. The NMN incubation also markedly raised the level of P65 and AC-P65 expression (Figure 6C). This effect was reversed by NMNAT1 overexpression, partially through consumption of excessive NMN. In contrast, NAMPT overexpression aggravated the effects of NMN as indicated by decreased Sirt1, but increased FN, P65, and AC-P65 levels. These findings confirmed that the accumulation of NMN under high glucose could be the cause of the increase in the cell fibrosis index involving Sirt1/NF-κB P65.

## 3. Discussion

Diabetic nephropathy is one of the leading causes of end-stage renal disease and creates heavy healthcare burdens worldwide. Since cellular decreased NAD^+^ concentrations and NAD^+^/NADH ratio are closely related to the pathogenesis of DN, the renal glomerular injury driven by NAD^+^ homeostasis imbalance has been thought to be a pathophysiological mechanism of DN [8,9]. NAD^+^ synthetase maintains the steady state of NAD^+^ homeostasis and plays a key role in the pathological development of DN, especially for regulating mesangial cells oxidative stress injury and extracellular matrix of glomerulosclerosis in DN [19,20,21]. NAMPT as a regulator of NAD^+^ biogenesis involved in cell energy metabolism systems and redox systems. However, exogenous extra cellular NAMPT has been reported to increase the synthesis of pro-fibrotic molecules in various types of renal cells [19,22]. As a type of extracellularly active inflammatory cytokine, NAMPT does not depend on the binding of lymphocyte antigen 96-TLR4, nor on other chaperones or cofactors such as lipopolysaccharide, but directly induces Toll-like receptor 4 (TLR4)-mediated NF-κB p65 activation [23]. Glomerular inflammation and fibrosis are likely induced by endogenous NAMPT overexpression [19]. Those findings suggested that suitable horizontal expression of endogenous NAMPT may be of importance for the physical state of the cell, whereas excess expression may promote pathological changes. Fibronectin is vital protein associated with the extracellular matrix (ECM), which has a significant role in the pathogenesis of inflammatory fibrosis of DN. Our present study demonstrated that endogenous NAMPT, FN, and NF-κB P65 expression increased significantly in mesangial cells and db/db diabetic mice renal tissues. Notably, Itaru Yasuda et al. [24] observed downregulated NAMPT expression and NMNAT1 expression at db/db mice 24 weeks of age. This difference may derive from different processing times, as stated by Yo Sasaki et al. [25]. The lack of an increase in NAD^+^ in response to the increased expression of endogenous NAMPT attributed to the fact that NMNAT1 downregulated reaction induced by HG, and thus decreased the NAD^+^ and NAD^+^/NADH ratio. Itaru Yasuda et al. [24] observed a decrease in the NMNAT1 level consistent with our results, and noticed mesangial expansion and foot process effacement present in the mice kidneys of DN. Furthermore, we found glomerulosclerosis in 20-week-old db/db DN mice and injury in mesangial cells incubated with 24 h HG paralleled decrease in NMNAT1, the NAD^+^ level, and NAD^+^/NADH ratio.

The oxidative stress level of cells in the diabetic environment is raised [26]. Reduced ROS production has been reported to evidently lessen mesangial dilation, glomerulosclerosis, and ECM protein accumulation including type IV collagen and FN in DM mice [8]. On the other hand, NADPH and GSH, as essential antioxidants in normal glucose metabolism, can prevent oxidative stress, but in hyperglycemia, NADP^+^/NADPH and GSSG/GSH are out of balance, resulting in reduced NADPH production and accumulation of H2O2, eventually inducing oxidative stress in the cells [27]. In this study, we verified that the level of ROS in HG-treated MSC was increased with DCFH-DA probe, and that NADPH and GSH decreased, but NADP and GSSG increased with LMSC. Similar results were reported in glomerular endothelial cells subjected to HG-mediated oxidative stress [28]. Consistent with these reports, overexpression of NMNAT1 raised NAD^+^ levels and reduced oxidative stress and ECM expression, while overexpression of NAMPT damaged cells in spite of the increased NAD^+^ level in our study. Thus, we proposed that the accumulation of NMN not only signify NAD^+^ anabolism disturbance in DN, but might have a direct regulatory effect on oxidative stress and extracellular matrix secretion. It has been reported that NMN accumulation led to axon injury [29], which can be alleviated upon NMN synthesis suppressed by FK866, a NAMPT inhibitor [25,29]. Our work shows that NMN accumulates in MSC during diabetic nephropathy, accompanied by a decrease in NAD^+^ levels, both of which are a disorder of anabolism. On the one hand, the decrease in NAD^+^ level can cause the increase in oxidative stress. On the other hand, the decrease in Sirt1 expression can increase cell injury through NF-κB P65/FN axis. However, although the expression of NAMPT can increase the level of NAD^+^ in MSC, the damage to the cells has not been alleviated. Therefore, it is reasonable to speculate that the metabolic disorder caused by NAD^+^ in high glucose environment, NMN accumulation, one of the manifestations of metabolic disorder of NAD, can also damage cells. Previous studies have shown that the accumulation of NMN promotes axonal denaturation when NMN is not consumed by enzymes. Excessive NMN can lead to increased consumption of NAD^+^ and degeneration of retinal cells by activating SARM1 [30,31]. Other researchers suggest that excess cytoplasmic NMN may interfere with the exchange of pyridine nucleotides between mitochondria and the cytoplasm, possibly resulting in mitochondrial dysfunction [32,33]. Previous studies have found that NMN inhibited NAMPT and exacerbated inflammation [34]. There may be a bond between NMN and NAP1L2, the latter being able to regulate the deacetylation function of Sirt1 [35]. We observed that the expression of FN increased with the increase in NMN concentration. Our results, however, do not prove that NMN accumulation is the cause of diabetic nephropathy, but rather that NAD^+^ metabolism is a disorder, whether NMN can bind alone or influence Sirt1 needs further study. Indeed, our data showed that HG increased NMN levels, and FK866 or overexpression of NMNAT1 prevented HG-accumulated NMN and decreased HG-induced damage in mesangial cells.

Sirt1 is a NAD^+^-dependent deacetylated enzyme that allows regulation of NF-κB P65, and plays a protective role in kidney [36]. Abnormal upregulation of endogenous NAMPT may lead to an imbalance between Sirt1 and NF-κB P65, and further aggravate cell damage that has been triggered by oxidative stress, eventually leading to cellular dysfunction and apoptosis [19]. In vitro study, we observed that FK866 or quercetin significantly reduced the expression of FN and NAMPT at the protein level, and suppressed the accumulation of NMN in mesangial cells cultured under high glucose condition. Moreover, the expression of Sirt1 and NMNAT1 was prominently enhanced by FK866 and quercetin. In addition, FK866 or quercetin also improved glycolysis in HG-injured mesangial cells. In vivo results further indicated that treatment with FK866 or quercetin improved renal function, reduced extracellular matrix deposition, and decreased glomerular sclerosis index; although, there was no significant improvement in body weight, kidney weight ratio, or blood glucose in db/db T2DM mice. More importantly, FK866 and quercetin consistently corrected NAD^+^ metabolic disorders, including expressions of NMNAT1, NAPRT, and NAMPT as well as NAD^+^ and NAD^+^/NADH levels both in vitro and in vivo.

In short, the results of this study provided the first evidence that NMN accumulation, possibly induced by up-regulation of NAMPT and down-regulation of NMNAT1 in diabetic nephropathy, plays a pathogenic role in DN with regulatory effects on NF-κB P65 and Sirt1 signaling pathways. Inhibiting NAMPT up-regulation and NMN accumulation might be the underlying mechanism of FK866 and quercetin, blocking the diabetic injury in vivo and in vitro. In conclusion, our findings suggested that prevention of NMN accumulation may be a promising target for DN treatment in future and warrants further study

## 4. Materials and Methods

### 4.1. Ethics Statement and Diagnosis

Permissions on performing animal experiments (No. 20170223–039) for research purposes were approved by the Ethics Committee of Shanghai Medical College, Fudan University, China. All procedures were carried out according to the approved guidelines. The definite diagnosis was made based on WHO histologic classification of glomerular diseases (1982 and 1995).

### 4.2. Cell Culture and Treatment

MSC (Human renal glomerular mesangial cells) were purchased from ScienCell Research Laboratories (Cat. #4200, ScienCell Research Laboratories, San Diego, CA, USA). MSC cultured according to the recommendatory protocols of ScienCell Research Laboratories. Normal glucose (NG) culture medium was mixed with high-glucose RPMI 1640 (Cat.2022400071, Thermo Gibco, Waltham, MA, USA) and non-glucose RPMI 1640 (Cat.118790, Thermo Gibco, Waltham, MA, USA) to end normal glucose containing 5.5 mmol/L d-glucose. High glucose (HG) culture medium was made by supplementing glucose into RPMI 1640 (Cat.2022400071, Thermo Gibco, Waltham, MA, USA) with additional d-glucose (Sinopharm, Shanghai, China) for a final d-glucose concentration at 30 mmol/L. NMN (N3501, Sigma-Aldrich, St. Louis, MA, USA) concentration gradient of MSC was incubated at 0, 0.005, 0.005, 0.05, and 0.5 (mM) for 24 h. After 24 h high glucose stimulated, MSC cultured and incubated with 50 nM FK866 (F85570, Sigma-Aldrich, St. Louis, MA, USA) or 20 μm quercetin (Q4951, Sigma-Aldrich, St. Louis, MA, USA) for 24 h. After treatment, MSC grown on 60 mm culture dishes (Corning, New York, NY, USA) were harvested and measured as depends.

### 4.3. Animal Studies

The 7- to 9-week-old male C57BL/6J and db/db mic were purchased from SLAC Laboratory (Shanghai SLAC Laboratory Animal, Shanghai, China) and maintained in accordance with Institutional Animal Care and used Committee procedures of Fudan University. Blood glucose of more than 15 mmol/L (280 mg/dL) were considered as diabetes. After 16 weeks, the db/db mice were then allocated randomly into db/db (n = 4), db/db with FK866 (n = 6), or db/db with quercetin (n = 6) groups. FK866 (20 mg/kg, biw, i.p.) or db/db with quercetin (100 mg/kg, biw, i.p.) was given for 4 weeks during the 16- to 20-week-old stage. All mice were maintained for 20 weeks and the blood glucose level and body weight were monitored every two weeks. At the end, 24 h urine was collected in the metabolic cages and mice were then sacrificed under chloral hydrate anesthesia to collect blood samples and kidney tissues. Total protein, creatinine, and albumin in urine, NAD^+^, and NADH in renal tissue were measured and analyzed according to the manufacturer’s instructions of standard diagnostic kits (Nanjing Jiancheng Bioengineering Institute, Nanjing, China).

### 4.4. Western Blot

Total proteins from MSC and animal renal tissues were extracted according to manufacturer’s protocols (Thermo Fisher Scientific, Waltham, MA, USA) and contained protease inhibitor cocktail (Selleck Chemicals, Houston, TX, USA). Using BCA protein assay kit (Thermo Pierce, Waltham, MA, USA), the protein concentration was determined. Proteins were loaded onto sodium dodecyl sulfate polyacrylamide gel electrophoresis (SDS-PAGE) (8–15%) and then transferred to polyvinylidene fluoride (PVDF) membranes (Millipore, Darmstadt, Germany). After blocking with 10% dried skim milk at room temperature for 1 h, the membranes were sheared and incubated with primary antibodies as follows: β-actin (1:10,000, 60009-1-lg, ProteinTech Group, Rosemont, IL, USA), NAPRT (1:1000, 13549-1-AP, ProteinTech Group, Rosemont, IL, USA), fibronectin (1:1000, ab45688, Abcam, Cambridge, MA, USA), NMNAT1 (1:500, sc-271557, Santa Cruz Biotechnology, Dallas, TX, USA), NAMPT (1:1000, 11776-1-AP, ProteinTech Group, Rosemont, IL, USA), acetyl-NF-κB P65 (1:1000, #12629, Cell Signaling Technology, Danvers, MA, USA), NF-κB P65 (1:1000, #8242, Cell Signaling Technology, Danvers, MA, USA), and Sirt1 (1:1000, 2977886, Millipore, Darmstadt, Germany) at 4 °C overnight. After incubated with horseradish peroxidase-conjugated antibodies at room temperature for 1 h, an enhanced chemiluminescence method from Millipore (Millipore, Darmstadt, Germany) was used and images were generated using a BIO-RAD Imaging System (BIO-RAD, Hercules, CA, USA). Bands of β-actin were used as an internal control.

### 4.5. Histology and Immunohistochemistry Analysis

The renal paraformaldehyde-fixed and paraffin-embedded sections were stained with hematoxylin–eosin (HE), Masson staining and Periodic Acid–Schiff (PAS), and glomerulosclerotic index was assessed on PAS staining. For immunochemistry experiments, 4 μm thick sections were fixed in 4% paraformaldehyde, blocked in 8% normal goat serum, and incubated in specific primary antibodies as follows: NAPRT (1:100), fibronectin (1:100), NMNAT1 (1:200), NAMPT (1:400), acetyl-NF-κB P65 (1:100), NF-κB P65 (1:100), and Sirt1 (1:200). After being washed three times by PBS, the sections were incubated with secondary antibody in 37 °C for 45 min. Hematoxylin was used as the nuclear counterstain and the expression was detected by diaminobenzidine. Images were taken with the Nikon camera (Nikon, Tokyo, Japan) and immunoreactivity was analyzed through Image J software (National Institutes of Health, Bethesda, MD, USA).

### 4.6. Transfection of Lentivirus

The plasmids that contained wild-type vector (PCDH-CMY-MCS-EF1-Puro) and constitutive overexpression cDNA (NAMPT or NMNAT1) were constructed by Genewiz (Genewiz, Suzhou, China) through amplification, purification, and sequencing (Appendix A). HEK293T cells (from Professor Qingquan Li, Department of Pathology at the School of Basic Medical Sciences, Fudan University) were cultured under standard conditions in the Dulbecco’s modified eagle’s medium (DMEM, Thermo Gibco, Waltham, MA, USA) supplemented with 10% bovine serum (Thermo Gibco, Waltham, MA, USA). For transfection, HEK293T cells were plated so that they reached 60–80% confluence and were transfected using Lipofectamine 2000 reagent (Invitrogen, Waltham, MA, USA). The virus solution was collected once every 24 h and 48 h, then filtered with a filter and mixed with 100 mg/mL Polybrene (Beyotime, Shanghai, China) at a ratio of 1:1000, and added to the culture solution of 50% target cells for 8–12 h and changed the fresh culture medium. The transfected MSC were screened with 1.5–2.5 μg/mL puromycin (Beyotime, Shanghai, China) after transfection for 48 h.

### 4.7. MSC Damage Detection

#### 4.7.1. Intracellular ROS Determination

After washed three times with PBS, to the MSC on 35 mm culture dishes (Corning, New York, NY, USA) was added 2,7-dichlorodihydrofluorescein diacetate (DCFH-DA, Beyotime, Shanghai, China) and incubated in the dark at 37 °C for 30 min. Images were taken using a fluorescence microscope (Leica, Wetzlar, Germany) and intracellular ROS levels analyzed by Image J software (National Institutes of Health, Bethesda, MD, USA).

#### 4.7.2. Cell Attachment Assay

MSC were grown in 35 mm culture dishes and then re-plating onto FN-coated surfaces. Cells were allowed to attach for 30 min, unattached cells were removed, and attached cells were fixed and counted. Imaged by a light microscope (Nikon, Tokyo, Japan) and analyzed through Image J software (National Institutes of Health, Bethesda, MD, USA).

### 4.8. Metabolite Collection and LCMS

After the collection of at least 1 × 10^5^ cell precipitates, and placed into 1 mL 80% methanol ice frozen at −80 °C one day ahead of time, the cells were repeatedly frozen in liquid nitrogen for 3 to 4 times, and then centrifuged for 10 min at 16,000× *g*, finally, supernatant was collected. The samples were entrusted to Institutes of Biomedical Sciences Fudan University for testing NAD^+^, NADH, NMN, NADP, NADPH, GSH, and GSSG.

### 4.9. Statistical Analysis

Data were expressed as mean ± SD. Differences between two and more groups were performed by means of the unpaired *t*-test or ANOVA, respectively, using Graph Pad Prism 7 (https://www.graphpad.com). The experiments were all independently repeated at least three times. * *p* < 0.05 was considered statistically significant.

## 5. Conclusions

In summary, we demonstrate in this study that FK866 and quercetin protects against DN through anti-fibrosis and antioxidant effects. The underlying mechanism, to our knowledge, is the first time to be found related to improved NAD^+^ anabolism with the prevention of intermediate NMN accumulation and promotion of product NAD^+^ synthesis. These improvements may further facilitate the renoprotecitve effects of NAD^+^-dependent deacetylase Sirt1 and its diverse downstream signaling pathways including NF-κB. Hence, targeting NAD^+^ anabolism disturbance characterized by NMN accumulation may pave the way to novel DN treatment.

## Figures and Tables

**Figure 1 ijms-23-03458-f001:**
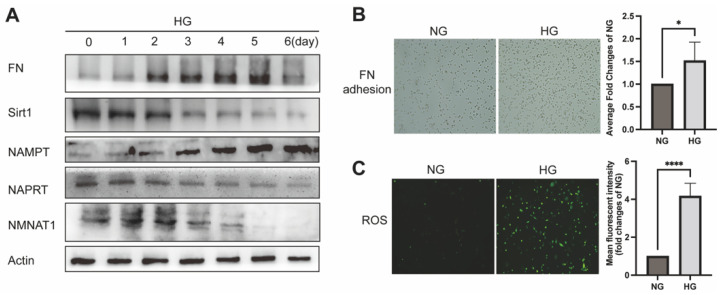
Effect of HG on NAD^+^-related enzymes and MSC injury: (**A**) Western blot analyzed NAD^+^-related enzymes and fibrotic injury of 0 to 6 days HG on MSC for FN, NAMPT, NAPRT, Sirt1, and NMNAT1 levels; β-actin was used as a loading control. (**B**) FN adhesion assayed adhesion and (**C**) DCFH-DA assayed ROS levels of MSC. 20 × 10 magnification; * *p* ≤ 0.01, **** *p* ≤ 0.0001; n = 3 independent samples; data represent mean ± SD; *t*-test analysis.

**Figure 2 ijms-23-03458-f002:**
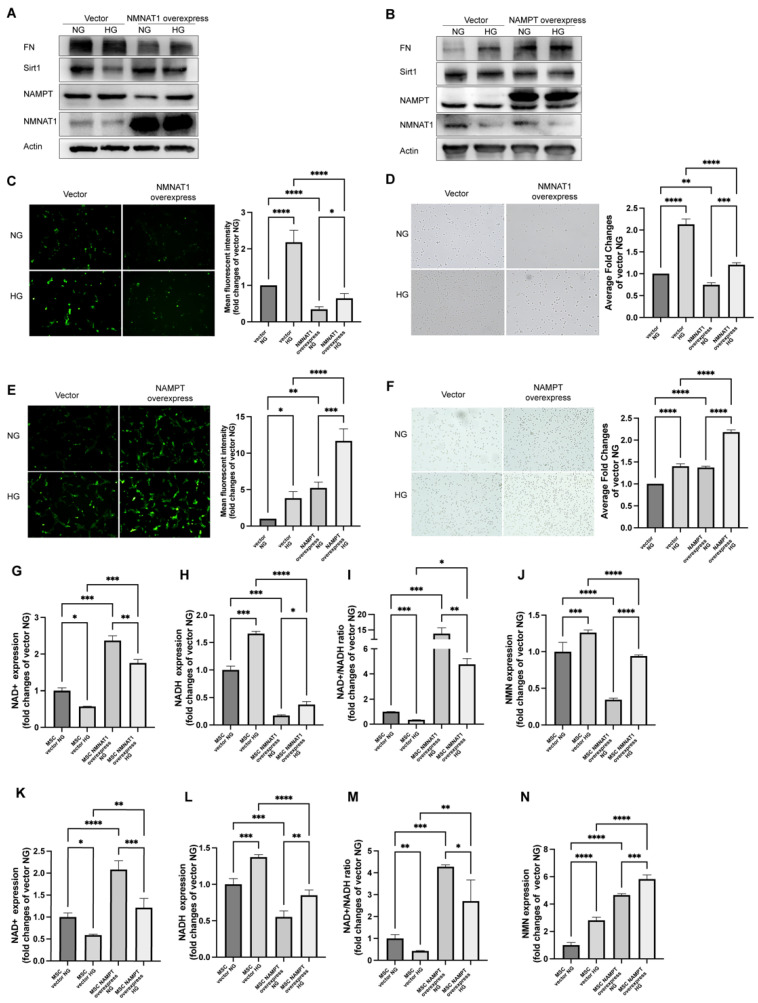
Effect of overexpression of NMNAT1 or NAMPT in MSC. Western blot analyzed NAD-related enzymes and fibrotic injury with or without 24 h HG in overexpressed NMNAT1 MSC (**A**) or overexpressed NAMPT MSC (**B**) for FN, Sirt1, NAMPT, and NMNAT1 levels; β-actin served as a loading control and shown were representative results from three experiments. DCFH-DA probe analyzed the ROS level with or without 24 h HG in overexpressed NMNAT1 MSC (**C**) or overexpressed NAMPT MSC (**D**); 20 × 10 magnification. FN adhesion assayed with or without 24 h HG in overexpressed NMNAT1 MSC (**E**) or overexpressed NAMPT MSC (**F**); 20 × 10 magnification. LCMS analysis of NAD^+^-related contents with or without 24 h HG in overexpressed NMNAT1 MSC (**G**–**J**) or overexpressed NAMPT MSC (**K**–**N**) for NAD^+^, NADH, NMN, and NAD^+^/NADH levels; n = 3 independent samples; data represent mean ± SD; * *p* ≤ 0.01, ** *p* < 0.01, *** *p* < 0.001, **** *p* < 0.0001.

**Figure 3 ijms-23-03458-f003:**
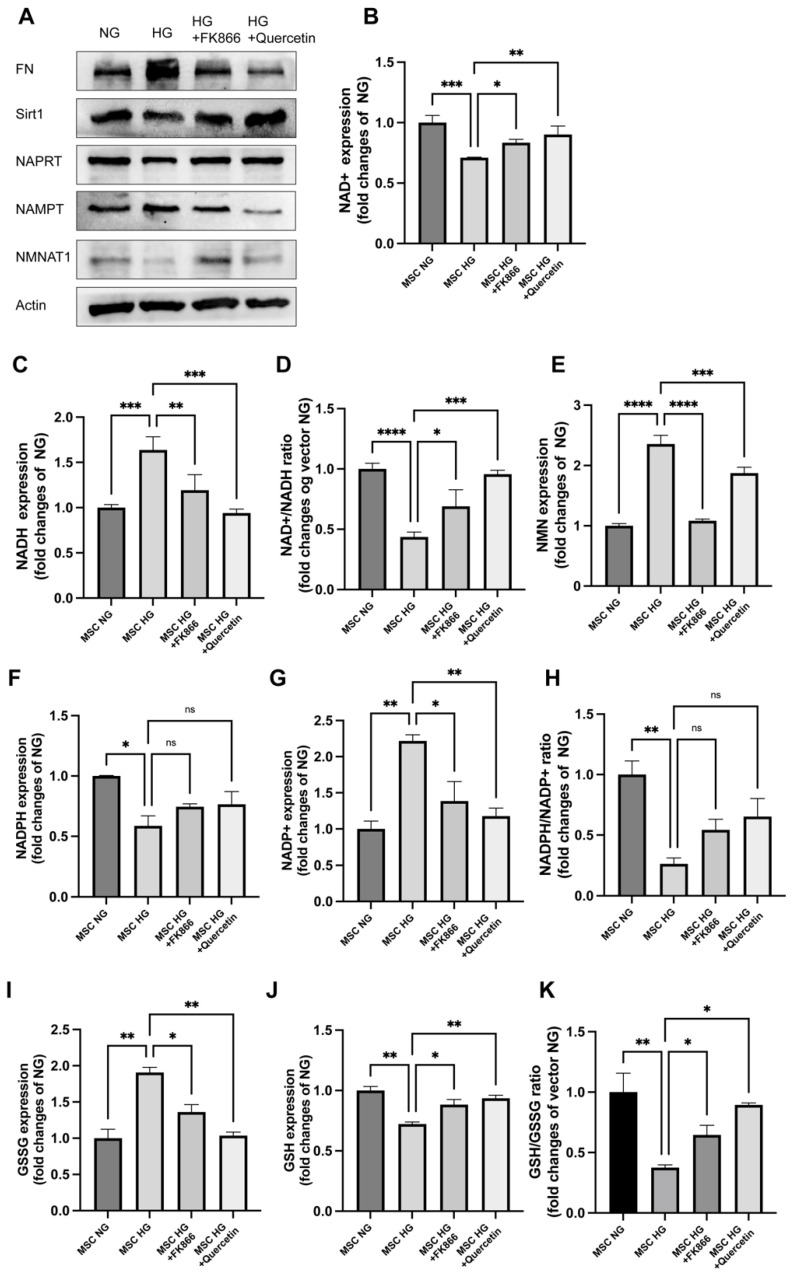
FK866 and quercetin protected against MSC injury and improved NAD^+^ anabolism under HG: (**A**) Western blot analysis of NAD-related enzymes and fibrotic injury incubated with FK866 (50 nM) or quercetin (20 μM) 24 h after 24 h HG in MSC for FN, NAMPT, NAPRT, and NMNAT1 expression; β-actin served as a loading control. LCMS analysis of NAD-related contents with FK866 (50 nM) or quercetin (20 μM) 24 h after 24 h HG in MSC for NAD^+^ (**B**), NADH (**C**), NAD^+^/NADH (**D**), and NMN (**E**); NADP^+^ (**F**), NADPH (**G**), and NADP^+^/NADPH (**H**); GSSG (**I**) GSH (**J**), and GSH/GSSG (**K**); n = 3 independent samples; data represent mean ± SD; ns means no significance, * *p* ≤ 0.01, ** *p* < 0.01, *** *p* < 0.001, **** *p* < 0.0001.

**Figure 4 ijms-23-03458-f004:**
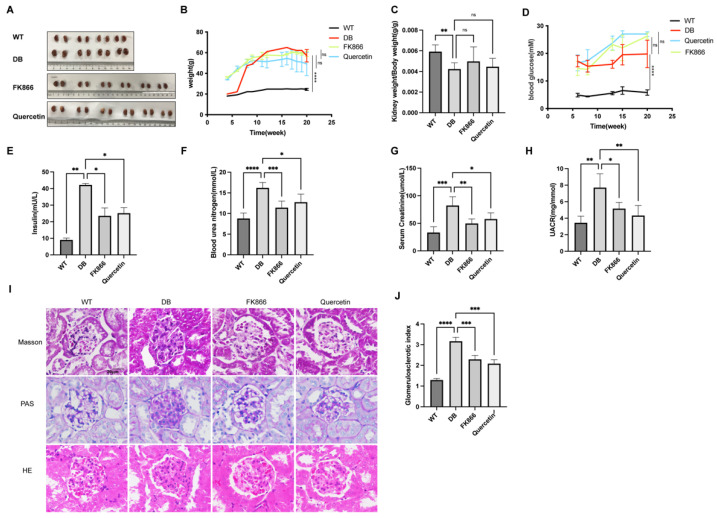
FK866 and quercetin effects on kidney function and renal pathology of morphology in db/db mice: (**A**) Renal images in different groups; WT or db/db group mice n = 4; FK866 or quercetin group n = 6. Measurement and analysis of the weight (**B**), kidney weight to body weight ratio (**C**), blood glucose (**D**), serum insulin (**E**) on mice. Renal function was evaluated by the levels of blood urea nitrogen (**F**), serum creatinine (**G**), and urine albumin to urine creatinine ratio (**H**). Light microscopy was used to observe renal morphologies. Representative photographs are shown for hematoxylin–eosin (HE), Masson and Periodic Acid–Schiff (PAS) staining with 20 × 10 magnification (**I**). The glomerulosclerotic index (**J**) were calculated from 5 to 10 randomly images of each mouse depending on PAS staining. Data are represented as the mean ± SD; n = 3; scale bar = 20 μm; ns means no significance, * *p* ≤ 0.01, ** *p* < 0.01, *** *p* < 0.001, **** *p* < 0.0001.

**Figure 5 ijms-23-03458-f005:**
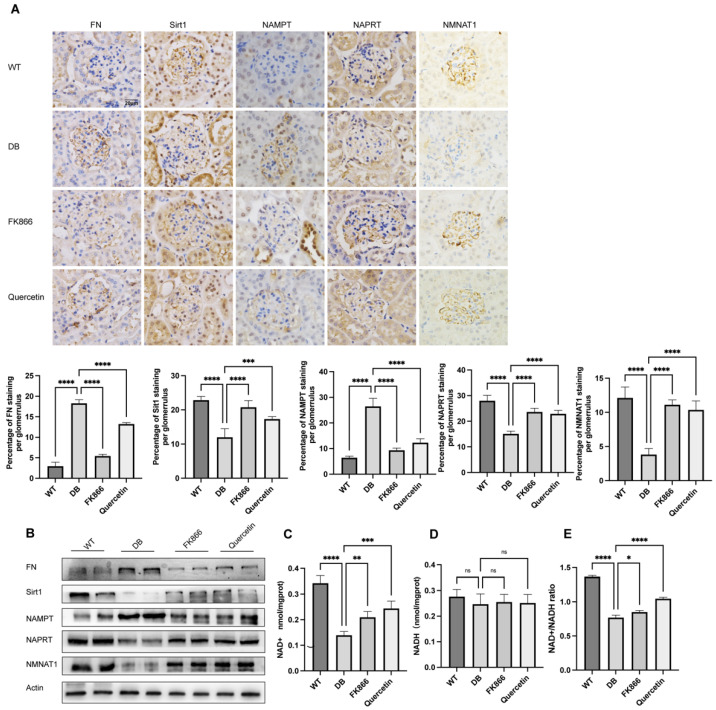
FK866 and quercetin effects on NAD^+^-related enzymes and fibrotic maker expression in mice: (**A**) Representative of immunohistochemical of mice renal tissues for FN, Sirt1, NAMPT, NAPRT, and NMNAT1; 20 × 10 magnification; scale bar = 20μm; data represent mean ± SD. Western blot analysis of FN, Sirt1, NAMPT, NAPRT, and NMNAT1 expression from renal tissues protein of mice (**B**). β-actin served as a loading control and the statistic based on the densitometric quantification of bands. NAD^+^ (**C**), NADH (**D**), and NAD^+^/NADH ratio (**E**) were measured in renal tissues from separated group mice; n = 3 independent samples; data represent mean ± SD; ns means no significance, * *p* ≤ 0.01, ** *p* < 0.01, *** *p* < 0.001, **** *p* < 0.0001.

**Figure 6 ijms-23-03458-f006:**
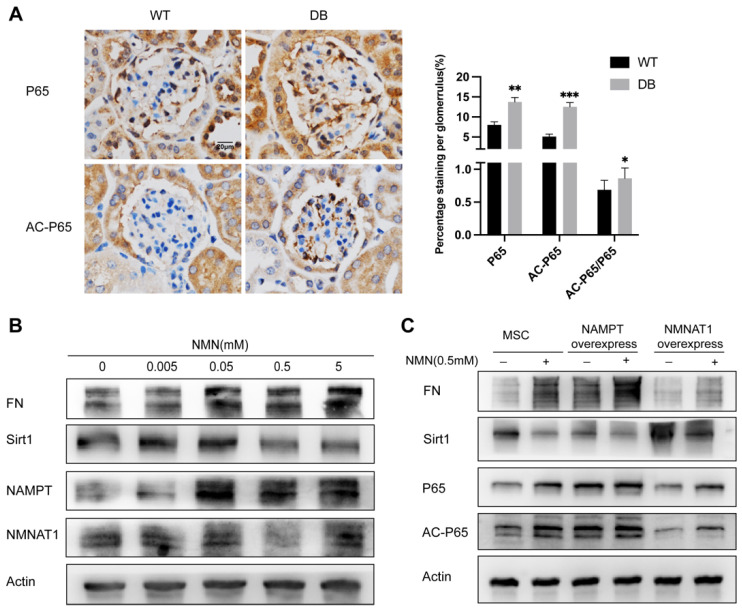
Effect of NMN on the Sirt1/NF-κB P65/FN signaling pathway in MSC: (**A**) Representative of immunohistochemical staining of mice renal tissues. The percentage of NF-κB P65 and ratio of acetylated P65/P65 stained area were calculated from 5 to 10 randomly images of each mouse. Scale bar = 20 μm. Western blot measured the dose effect of NMN on FN, Sirt1, NAMPT, and NMNAT1 expressions at an increasing dose gradient in MSC using (**B**) and measured Sirt1, FN, NF-κB P65, and acetylated P65 (AC-P65) incubated with or without NMN (0.5 mM) for MSC or transfected MSC, using β-actin as a control (**C**). * *p* ≤ 0.01, ** *p* < 0.01, *** *p* < 0.001.

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
