# Peer review of "NAD+ Anabolism Disturbance Causes Glomerular Mesangial Cell Injury in Diabetic Nephropathy"

_ijms, 2022, doi:10.3390/ijms23073458_

Round 1

Reviewer 1 Report

The work of Xue Li et al explores the homeostasis of NAD+ anabolism in relation to the development of diabetic nephropathy with particular interest in the accumulation of nicotinamide mononuclotide (NMN).  Their results are apparently convincing in establishing a link between NMN accumulation and NAD+ anabolism disturbance and diabetic nephropathy (DN) indicating this mechanism  as a promising therapeutic target for DN treatment. Indeed Quercetin protects mesangial cells from oxidative damage. The experiments are complete both in vivo and in vitro and from the phenomenological point of view suggest a role of NMN in the development of diabetic nephropathy and how this process can be at least improved with quercetin.

The experiments are well conducted and exhaustive both in vitro and in vivo, but the problem remains of determining what the true role of NMS is in the development of diabetic nephropathy. The discussion should be more probabilistic without the possibility of identifying the mechanism described as a cause of diabetic nephropathy.

Reviewer 2 Report

In this interesting research paper, the authors aimed to investigate the role of NAD+ Sirt1/NF-κB P65 pathway at the glomerular mesangial cell level in Diabetic Nephropathy.

The topic is of great interest in the scientific community and well described. The experimental methods are described comprehensively, the statistical analysis is adequate, and the conclusions are in line with previously published research.

I support the publication of this paper after several improvements.

In the introduction section, I suggest you include the role of therapeutic inertia in Diabetic Kidney Disease (See the paper Kidney Disease in Diabetic Patients: From Pathophysiology to Pharmacological Aspects with a Focus on Therapeutic Inertia. Int J Mol Sci. 2021 May 1;22(9):4824. doi: 10.3390/ijms22094824. PMID: 34062938; PMCID: PMC8124790).

Please put the Material and Methods paragraph before the results paragraph.

It is of pivotal importance that you add the conclusion section.

You should amend your references so that it adheres to the journal style format “Journal references must cite the full title of the paper, page range or article number, and digital object identifier (DOI) where available. Cited journals should be abbreviated according to ISO 4 rules”. 

Round 2

Reviewer 1 Report

the modifications made by the authors satisfy the requests and the work can be published